# Field Position-Related Variations in Body Mass, Postural Control, and Isokinetic Strength in Portuguese Professional Football

**DOI:** 10.3390/jfmk10040447

**Published:** 2025-11-19

**Authors:** João Rocha, Hélder Cruz, José Eduardo Teixeira, Carolina Vila-Chã, Raúl Filipe Bartolomeu, João Nuno Ribeiro, Faber Martins, Pedro Tiago Esteves

**Affiliations:** 1Department of Sports and Expressions, Polytechnic of Guarda, 6300-559 Guarda, Portugal; joaopdsrocha@gmail.com (J.R.); hbdcruz17@gmail.com (H.C.); jose.eduardo@ipg.pt (J.E.T.); bartolomeu@ipg.pt (R.F.B.); jnribeiro@ipg.pt (J.N.R.); fabermartins@ipg.pt (F.M.); 2Sport Physical Activity and Health & Innovation Center (SPRINT), 2040-413 Rio Maior, Portugal; cvila-cha@ipca.pt; 3Department of Sports Sciences, Polytechnic of Cávado and Ave, 4806-909 Guimarães, Portugal; 4Research Centre for Active Living and Wellbeing (LiveWell), Instituto Politécnico de Bragança, 5300-253 Bragança, Portugal; 5Department of Sport Sciences, Instituto Politécnico de Bragança, 5300-253 Bragança, Portugal; 6Department of Sport Sciences, University of Beira Interior, 6201-001 Covilhã, Portugal; 7Research Centre in Sport Sciences, Health Sciences and Human Development, CIDESD, Creative Lab Research Community, 5000-801 Vila Real, Portugal; 8Portugal Football School, Portuguese Football Federation, 1495-433 Oeiras, Portugal

**Keywords:** strength, dynamic balance, body composition, positional, football

## Abstract

**Background**: Combining body composition, dynamic postural control, and isokinetic strength allows for a comprehensive physical and functional assessment of football players across specific playing positions. The aim of this study was to analyze the field position-related variations in the physical and functional profiles of male football players. **Methods**: A convenience sample of 23 professional male football players from a Portuguese second league team, aged 18 to 32 years (23.83 ± 3.77 years), participated in the present study. Players from five field positions (central backs, central midfielders, wide midfielders, and forwards) were assessed for body composition, dynamic postural control, and isokinetic lower limb strength. Body composition was assessed by bioelectrical impedance; composite scores for postural control in the right and left limbs were obtained through the Y-Balance test lower quarter (YBT-LQ). Peak torque (PT) during lower limb flexion and extension was measured using an isokinetic dynamometer chair. **Results**: Significant differences were found only in peak torque of the right extensors [H (4) = 9.84, *p* = 0.043, η^2^ = 0.37)], whereas no significant differences emerged in body mass, dynamic balance scores, left-side extension and flexion, or hamstring-to-quadriceps (H:Q) strength ratios. Post hoc analysis using Mann–Whitney U tests with Holm correction did not confirm pairwise differences between positions. The correlation analysis revealed mainly moderate-to-strong associations between symmetrical measures (composite YBT-LQ right and left, PT extension right and left), as well as between anthropometric and strength variables, but without consistent position-specific patterns. **Conclusions**: Overall, the study provides valuable insights into the physical attributes of professional football players, highlighting the general similarity of profiles across positions and suggesting that individualized training approaches may still be beneficial to optimizing performance and reduce injury risk. Future studies should extend the analysis to larger cohorts across different professional Portuguese football teams and competitions.

## 1. Introduction

Football is a multifaceted sport in which the game demands vary according to the player’s specific positioning. To perform effectively, players are obliged to adapt their technical and tactical skills, along with physical propensities, to the challenges related with specific playing position [1]. Within such a demanding competitive environment such as football game, players face abrupt oscillations between high-intensity periods of exertion and low-intensity actions, where a complex array of information is readily available to be perceived and acted upon [2]. Of interest is the fact that the evolution of the football game, from the 90s to the present, has signaled an increasing importance attributed to the capacity to perform high-speed actions that require an ability to activate anaerobic metabolism along with the correspondent muscular structure [3].

Body composition and lower limb muscle performance are key factors that directly influence the performance and longevity of professional players [4]. Body composition is a critical determinant of performance in football, affecting agility, speed, and endurance. Conversely, excessive fat mass can impair both aerobic and anaerobic performance, potentially compromising overall athletic capacity [5]. Research has consistently shown the existence of field-position-related variations in terms of body composition. For instance, Sebastiá-Rico et al. [6] reported that midfielders stand as the playing position with the lowest height, total height, and muscle mass. Curiously, defenders, together with goalkeepers, stand out with the greatest height, sum of skinfold, and muscle mass.

Postural control, both static and dynamic, is crucial for maintaining stability during high-intensity actions such as changes of direction and jumps. Several researchers suggest that balance demands vary by position, with midfielders and forwards requiring greater balance due to their frequent and rapid movements compared to defenders [7]. In this context, the Y-Balance Test (YBT) has been recognized as one of the most popular tests to assess dynamic postural control due to its practicality and low-cost. This test requires athletes to balance one leg while reaching as far as possible with the other leg in three distinct directions: anterior, posterolateral, and posteromedial.

Usually, for professional footballers, composite scores typically range around 102%, with specific values indicating strong bilateral symmetry in balance performance. For example, Butler et al. observed an average YBT composite score of 102% among 44 professional players, indicating comparable proficiency within this group [8]. Moreover, Tekin et al. [9] revealed mediolateral stability indices for dynamic balance that indicate high functional abilities in football players, with dynamic balance scores averaging approximately 8.32 ± 4.98. Research has supported the correlation between YBT performance and injury prediction, indicating that lower composite scores, often less than 14, are predictive of higher injury risks [8,10]. Consequently, the YBT provides a comprehensive assessment of an athlete’s strength, stability, and balance in multiple movement planes [11]. Specifically, the Y-Balance Test Lower Quarter (YBT-LQ) can inform clinical practice by supporting targeted interventions to enhance dynamic postural control, mobility, and hamstring strength, with a particular focus on strengthening the hamstrings and addressing deficits in movement mechanics [11]. Increased joint angles in knee flexion, ankle dorsiflexion, and trunk ipsilateral flexion have emerged as key contributors to anterior reach performance [7]. Furthermore, the associated kinematic patterns suggest a specific motor strategy adopted by healthy individuals to optimize reach during the YBT-LQ [7,8].

The isokinetic test is a widely used method for assessing lower limb muscle strength with valuable insights into physical performance in professional football players [12,13]. Knee flexor and extensor muscle strength, along with the hamstring–quadriceps ratio, have been identified as key factors in assessing the risk of lower extremity injuries [14,15,16]. Furthermore, significant differences in physical constitution and weight between players in different field positions can have a significant influence on the level of isokinetic strength. For instance, in a cohort of 111 elite international polish players, examined for 5 years, it was identified that goalkeepers and central midfielders presented lower peak torque (for quadriceps and hamstrings) compared with other playing positions. In addition, a tendency for the dominant leg to present higher isokinetic performance than the non-dominant leg was also found [17]. Also, Parpa and Michaelides [18] reported that concentric torque measurements at a speed of 60 degrees per second for the quadriceps were between 201 and 319 N·m, while the hamstrings ranged from 114 to 170 N·m. In fact, there is solid experimental evidence indicating that isokinetic strength performance varies significantly among players in different playing positions.

The present study examined the relationship between isokinetic data, body composition, and postural control performance, in line with the previous body of research in this domain [6,8,14]. While most of the previous studies analyzed these variables separately, the interplay between body composition, dynamic postural control, and isokinetic strength across different playing positions remains underexplored in professional Portuguese football. Understanding these factors is essential for coaches, sports scientists, and medical staff to optimize training programs, prevent injuries, and enhance performance based on the specific demands of each playing position. Thus, the aim of this study was to analyze the field position-related variations in the physical and functional profiles of male football players from a Portuguese second league team. By investigating the interplay between these variables, it was hypothesized that players in different field positions would display distinct physical and functional profiles, particularly in muscle strength and postural control parameters, reflecting the specific physiological and biomechanical demands of their playing roles.

## 2. Materials and Methods

### 2.1. Subject

A descriptive, comparative and observational cross-sectional study was conducted with a convenience sample of 23 male professional football players from a Portuguese second league team. The participants were aged between 18 and 32 years (23.83 ± 3.77), with an average body weight of 79.21 ± 6.7 kg and an average height of 184.54 ± 6.6 cm. The participants were aged between 18 and 32 years (23.83 ± 3.77), with an average body mass of 79.21 ± 6.70 kg and an average height of 184.54 ± 6.60 cm. The sample was categorized by field positions, specifically: center backs (CBs, n = 4), full-backs (FBs, n = 4), central midfielders (CMs, n = 5), wide midfielders (WMs, n = 4), and forwards (FWs, n = 4) (Table 1). The study was carried out during the team’s pre-season, using a non-probabilistic sampling method.

The data were collected as part of routine preseason performance assessments, which are mandatory within the players’ professional duties and constitute standard practice rather than an experimental protocol. Consequently, no formal request for ethical committee approval was required. All participants were fully informed about the objectives and potential risks associated with the study and provided written consent for the anonymized use of their data for academic and research purposes.

### 2.2. Selection Criteria

The research inclusion criteria were as follows: (i) male professional football players from a Portuguese second-league team; (ii) participation in the pre-season phase of training; (iii) completion of full assessment protocol; and (iv) provision of informed consent and agreement to participate in all study procedures. The exclusion criteria for the study were as follows: (i) have any musculoskeletal injury in the last 6 months; (ii) a history of recent surgery or medical conditions affecting body composition, dynamic postural control, or muscle strength; (iii) inability to complete the YBT test, isokinetic strength test, or anthropometric measurements as required; (iv) use of performance-enhancing drugs or supplements that could influence body composition or physical performance; and (v) refusal to provide informed consent or to comply with study procedures.

### 2.3. Instruments and Procedures

#### 2.3.1. Body Composition

Weight, height, and body mass were analyzed based on anthropometric recommendations [4]. Also, the lower limb height was collected for the calculation of relative ratios in the YBT. First, measurements of both lower limbs were taken from the iliac crest to the ankle, and the dominant foot of each athlete was notated. The athletes began by removing their shoes and lying supine on a massage table for the limb measurements. Then, and still barefoot, their height was measured in centimeters using an anthropometer, with the athlete positioned in an anthropometric position and their head oriented in the Frankfurt position. Finally, body composition was determined using the bioimpedance scale InBody 270^®^ (Inbody Co. Ltd., Seoul, South Korea), following the manufacturer’s standard procedure.

#### 2.3.2. Dynamic Postural Control

After determining their body composition, the players performed the YBT test using the Perform Better YBT Kit^TM^. Only the YBT-LQ was collected. Prior to testing, they were instructed to perform a light warm-up on an exercise bike. Next, all Y-Balance test procedures and body positioning were explained to the participants, as described elsewhere [7]. The test was performed in the following order, for each foot: anterior, posteromedial, and posterolateral. Adequate recovery time between evaluations was ensured. The sum of the three normalized reach distances was averaged and multiplied by 100 to generate a composite score (CS). Additionally, the absolute reach difference between lower limbs was calculated to assess reach symmetry [19,20].

#### 2.3.3. Isokinetic Muscle Strength

Finally, the isokinetic strength test for lower limbs was performed using an isokinetic dynamometer (Biodex System 4 Pro, Shirley, NY, USA). Before the evaluation, athletes performed a general activation on a stationary bike for 10 min at medium intensity [21]. Then, the athletes were seated in the dynamometer chair, and their position was stabilized using belts placed at the torso, abdomen, and thigh to prevent accessory movements. The knee to be evaluated was positioned at 90 flexion (0° = full extension), and the dynamometer lever arm was aligned with the lateral femoral condyle. After the positioning and alignment procedures, the athletes performed several submaximal flexion and extension movements to complete the muscle activation period and familiarize themselves with the equipment and testing procedures [14,22]. During the evaluation, the angular speed used for isokinetic tests was 60°/s for concentric strength of knee extensor and flexor muscles. All data were recorded using the Biodex™ Advantage BX software (Biodex Medical Systems, Shirley, NY, USA) and later exported to a spreadsheet (Excel, Microsoft, Redmond, WA, USA) for further analyses. The peak torque—right/left knee extension, PT_extension_ (R) and PT_extension_ (L), represents the maximum torque (force) generated by the extensor muscles of the right and left knee (primarily the quadriceps) during the isokinetic test. It is measured in units of torque (N·m) and reflects the strength of the right and left knee extensors. The peak torque—right/left knee flexion, PT_flexion_ (R) and PT_flexion_ (L), represents the maximum torque generated by the flexor muscles of the right and left knee (primarily the hamstrings) during the isokinetic test. It represents the strength of the right and left knee flexors. The hamstring-to-quadriceps ratio—right and left leg, ratio H:Q (R) and ratio H:Q (L), portray the ratio between the peak torque produced by the hamstrings (knee flexors) and the quadriceps (knee extensors) of the right leg. It provides insight into muscle balance around the knee joint, with an ideal ratio typically ranging between 50–80% depending on the context: injury history, dynamometer type, period of the season testing has been performed and competition level; training is also a contextual factor affecting isokinetic testing outcome [23,24].

### 2.4. Statistical Analysis

Descriptive statistics were presented as means, standard deviations (Mean ± SD), and coefficients of variation (CV). Group comparisons between playing positions were performed using the non-parametric Kruskal–Wallis test, with post hoc pairwise comparisons. A priori power analysis was performed in G*Power v. 3.1.9.7 (F tests, ANOVA: fixed effects, one-way), using the standard approximation for the Kruskal–Wallis test. Assuming *k* = … groups, *α* = 0.05, power 1 − *β* = 0.80. For H statistics, the overall effect size was estimated by the rank-based eta-squared (η^2^_H), calculated from the H statistic of the test. The proportion of variance in ranks can be explained by differences between groups and is interpreted according to the following reference values: 0.01 (small), 0.06 (moderate), and 0.14 (large). Correlation analysis was conducted with Spearman’s rank correlation coefficient (ρ) and corresponding *p*-values. The magnitude of correlation was classified as: trivial if |ρ| ≤ 0.1, small if 0.1 < |ρ| ≤ 0.29, moderate if 0.3 ≤ |ρ| ≤ 0.49, large if 0.5 ≤ |ρ| ≤ 0.69, very large if 0.7 ≤ |ρ| ≤ 0.89, and almost perfect if |ρ| ≥ 0.9 [25]. The CV (%) was calculated as the ratio between the standard deviation and the mean, multiplied by 100, to express the relative variability of the data. Statistical analyses were performed with a significance level set at 5% using JASP (version 0.16.3.0, JASP, The Netherlands, https://jasp-stats.org/) and Python (version 3.13.5, Python Software Foundation, https://www.python.org/). Data tables and variable organization were performed using SPSS (version 23, IBM, USA).

## 3. Results

Descriptive statistics for body composition, dynamic postural control (right and lefts CS), and isokinetic muscle strength are summarized (Table 2). The CV ranged from 0.221 to 0.028.

Table 3 presents the results of the non-parametric Kruskal–Wallis analysis comparing field positions across anthropometric, dynamic postural control, and isokinetic muscle strength variables. No statistically significant differences were observed between field positions for age (H = 9.00, *p* = 0.061, η^2^ = 0.31), height (H = 3.21, *p* = 0.523, η^2^ = 0.01), body mass (H = 7.83, *p* = 0.098, η^2^ = 0.24), composite YBT right (H = 4.67, *p* = 0.323, η^2^ = 0.04) and left (H = 5.61, *p* = 0.231, η^2^ = 0.10), PT extension left (H = 6.68, *p* = 0.154, η^2^ = 0.17), PT flexion right (H = 2.85, *p* = 0.583, η^2^ = 0.01), and PT flexion left (H = 1.77, *p* = 0.778, η^2^ = 0.03). A statistically significant difference was detected for PT extension right (H = 9.84, *p* = 0.043, η^2^ = 0.37), indicating a large effect size. Although several positional trends were evident in the descriptive data, the results revealed that most group differences did not reach statistical significance. Nonetheless, pairwise post hoc comparisons identified significant differences between FB and CM (*p* = 0.029) and between FB and FW (*p* = 0.004), suggesting position-specific variations in right knee extensor performance.

Figure 1 shows the Spearman’s rank correlation analysis revealed several significant associations between anthropometric characteristics and isokinetic performance variables for each position and overall sample. A strong positive correlation was observed between body mass and height (ρ = 0.80, 95% CI [0.56, 0.92], *p* < 0.001). Likewise, body mass correlated positively with PT extension of both limbs (right: ρ = 0.61, 95% CI [0.26, 0.82], *p* = 0.002; left: ρ = 0.43, 95% CI [0.02, 0.72], *p* = 0.04) and with PT flexion (right: ρ = 0.73, 95% CI [0.44, 0.88], *p* < 0.001). Height demonstrated moderate-to-large positive associations with right extensor torque (ρ = 0.43, 95% CI [0.02, 0.72], *p* = 0.04) and right flexor torque (ρ = 0.60, 95% CI [0.24, 0.82], *p* < 0.01), supporting a linear trend between anthropometric size and torque generation capacity.

A strong association was also found between right and left composite YBT scores (ρ = 0.89, 95% CI [0.74, 0.95], *p* < 0.001), confirming measurement consistency and inter-limb coordination in dynamic balance. Regarding muscular balance, the H:Q ratio (right) correlated negatively with right extensor torque (ρ = −0.53, 95% CI [–0.78, –0.14], *p* = 0.01) but positively with right flexor torque (ρ = 0.56, 95% CI [0.18, 0.80], *p* = 0.006), reflecting the expected inverse relationship between absolute torque magnitude and proportional balance ratios. The left H:Q ratio correlated positively with left flexor torque (ρ = 0.45, 95% CI [0.04, 0.74], *p* = 0.03) and strongly with the right H:Q ratio (ρ = 0.67, 95% CI [0.36, 0.84], *p* < 0.001), suggesting symmetrical neuromuscular control between limbs. No significant correlations were observed between age and any torque or H:Q ratio variables (*p* > 0.05).

When analyzed by position, central backs (CBs) showed strong positive correlations between height, body mass, and both extensor and flexor torque (ρ = 0.43–0.73, *p* ≤ 0.05), alongside a negative association between H:Q (right) and PT extension (ρ = −0.53, *p* = 0.01). Full-backs (FBs) exhibited moderate correlations between body mass and PT extension (ρ = 0.43–0.61, *p* < 0.05) and between height and flexor torque (ρ = 0.60, *p* = 0.002), as well as a positive association between H:Q ratios and PT flexion (ρ = 0.56, *p* = 0.006). Central midfielders (CMs) demonstrated moderate but non-significant correlations among most torque variables (ρ = 0.19–0.27, *p* > 0.05) and a strong bilateral association between YBT composite scores (ρ = 0.89, *p* < 0.001). Wide midfielders (WMs) presented positive associations between H:Q ratios and flexor torque (ρ = 0.45–0.56, *p* < 0.05). Forwards (FWs) displayed strong positive correlations between flexor torque and both height and body mass (ρ = 0.60–0.73, *p* < 0.01), as well as between H:Q (right) and (left) ratios (ρ = 0.67, *p* < 0.001), indicating consistent bilateral muscle balance patterns.

Regarding muscular balance, the H:Q ratio (R) correlated negatively with right extensor torque (ρ = −0.53, *p* = 0.010) but positively with right flexor torque (ρ = 0.56, *p* = 0.006), reflecting the expected inverse association between torque magnitude and ratio proportion. The left H:Q ratio correlated positively with left flexor torque (ρ = 0.45, *p* = 0.032) and strongly with H:Q (R) (ρ = 0.67, *p* < 0.001), suggesting bilateral symmetry in muscle balance. No significant correlations were found between age and any torque or ratio variables (*p* > 0.05).

Considering the positional differences, CB players showed strong positive correlations between height, body mass, and extensor/flexor torque (ρ = 0.43–0.73, *p* ≤ 0.05), and a negative correlation between H:Q (R) ratio and PT _extension_ (ρ = −0.53, *p* = 0.010). FB players presented moderate correlations between body mass and PT _extension_ (ρ = 0.43–0.61, *p* < 0.05) and between height and flexor torque (ρ = 0.60, *p* = 0.002), with a positive association between H:Q ratios and PT _flexion_ (ρ = 0.56, *p* = 0.006). CMs showed moderate and non-significant correlations among most torque variables (ρ = 0.19–0.27, *p* > 0.05) and a high bilateral association between composite YBT scores (ρ = 0.89, *p* < 0.001). WMs showed positive correlations between H:Q ratios and flexor torque (ρ = 0.45–0.56, *p* < 0.05). FW players demonstrated positive correlations between flexor torque and both height and body mass (ρ = 0.60–0.73, *p* < 0.01), and between H:Q (R) and (L) (ρ = 0.67, *p* < 0.001).

## 4. Discussion

This study aimed to analyze body composition, dynamic postural control, and isokinetic lower limb strength in football players from a Portuguese second league men’s team. We targeted the existing gap related to the interplay between these variables and football playing positions. Our main findings disclosed significant variations in body mass and right-side knee flexion torque across playing positions, whereas dynamic postural control and other strength measures remained consistent.

The study identified that significant differences were detected only for PT_extension_ (R). Although some positional trends were evident in the descriptive statistics, significance annotations have been described for FB vs. CM, and FB vs. FW. This highlights the importance of tailored training for each position [6]. However, no significant differences were found for other variables, including dynamic postural control and hamstring-to-quadriceps strength ratios [14,23]. These findings underscore the influence of position-specific demands on physical attributes, emphasizing the need for individualized training programs to optimize performance. Midfielders are typically required to sustain high levels of endurance and mobility due to the nature of their role, which often involves covering large areas of the pitch, linking defense with attack, and maintaining a high tempo throughout the match [25]. Consequently, their training programs emphasize aerobic conditioning, agility, and stability to enhance postural control, thereby enabling efficient movement transitions and balance maintenance during rapid directional changes. This need for superior endurance and agility may lead to adaptations in body mass that favor stamina and efficient movement, as well as improvements in neuromuscular control that support dynamic postural stability [26,27]. In contrast, forwards are generally tasked with executing explosive actions such as rapid sprints, powerful shots, and sudden changes of direction in confined spaces. These tactical requirements necessitate a training focus on explosive power and high-intensity strength, which often results in the development of greater muscle mass and higher isokinetic strength, particularly in the lower extremities [28].

Overall, the results suggest that the players presented relatively consistent profiles across positions in terms of body composition, postural control, and muscle strength, except for PT extension (R). In comparison to other studies, our findings are in contrast with existing research that highlights the influence of positional demands on physical attributes such as body mass and muscle strength [29]. However, the absence of significant differences in dynamic balance also contrasts with previous evidence suggesting that balance demands are higher for midfielders and forwards due to their frequent and rapid changes in movement [23,30]. Furthermore, full-backs showed significant differences compared to central midfielders and forwards, likely due to their unique spatial and physical demands during matches that involve repetitive overlapping runs and defensive duties [1,28]. Interestingly, there were no similar differences between full-backs and wide midfielders or wingers, which could be explained by the similarity in roles that require rapid directional changes and sustained lateral movement and tasks that may lead to overlapping muscular adaptations [2]. The distinct profile of full-backs underscores the importance of position-specific training to optimize performance and minimize injury risk, especially given their unique movement patterns and physical demands compared to other outfield roles. These results should consider that methodological differences, including variations in balance assessment protocols and sample characteristics, may account for these discrepancies. Similarly, while this study did not observe significant differences in hamstring-to-quadriceps ratios, previous research has associated imbalances in these ratios with an increased risk of injury [10,14]. Furthermore, the measures directly related to positional-specific demands such as body mass and power output, no significant differences were found in variables like dynamic postural control and H:Q strength ratios [30]. This convergence may be explained by standardized training protocols, recovery protocols, or overall physical conditioning programs employed within the Portuguese professional football context that aim to maintain a balanced neuromuscular performance across various playing positions. It is also possible that certain aspects of dynamic balance are universally emphasized due to the collective demands of professional football, regardless of positional role [31]. The literature shows that differences related to physical profile, postural control, and muscle strength between different positions in football are limited or vary depending on the level of athlete specialization [4,6]. Moreover, studies indicate that, in most cases, players exhibit relatively homogeneous profiles in these variables, reflecting an adaptation to the common demands of training and competition, regardless of the specific position [1,5]. However, the exception of PT extension (R) may be related to the particular physical and biomechanical requirements of certain roles on the field, which demand greater or lesser specific muscle activation [2,12]. Studies assessing differences between positions also highlight that body composition, postural control, and muscle strength can vary according to factors such as playing level, age, and tactical role [18,28]. Therefore, the observed uniformity suggests that, despite functional differences, there is a common pattern of organic adaptation in high-level football, with the specific exception of PT _extension_, which may reflect a functional characteristic or injury risk associated with the position or role played [10,23]. Hence, the finding of relatively consistent profiles between positions, except for PT _extension_ (R), aligns with the literature that recognizes both overall homogeneity and functional particularities of each role on the field [25,32].

Previous research has associated imbalances in H:Q strength ratios with an increased injury risk of muscle strain and other injuries, emphasizing the need for balanced strength training [22]. However, this study did not observe significant differences in hamstring-to-quadriceps ratios. The absence of significant differences might be due to the role of training programs, recovery protocols, or the specific demands of the Portuguese second league. This signals the need for the ongoing monitoring and adjustment of training programs to prevent injuries and optimize performance. It is possible that the training regimens in the Portuguese second league are effective in maintaining balanced strength ratios, which could explain the lack of significant differences observed. Considering muscular function, particularly regarding the quadriceps and hamstrings, the roles of agonists and antagonists are crucial in understanding positional differences observed in PT_extension_ (R). The quadriceps serve as the primary agonists during knee extension, while the hamstrings act as antagonists, contributing to joint stability and dynamic movement control [22,23]. Optimal ratio ranges are typically 50–80% but vary by angular velocity and contraction type. Also, bilateral asymmetry thresholds >10–15% are often considered clinically significant.

The significant difference detected solely for PT_extension_ (R) could indicate positional-specific adaptations in muscle strength and balance, potentially reflecting different demands placed on these muscle groups depending on playing role. Variations in quadriceps strength, which influence extension capacity, may be associated with the tactical and biomechanical requirements of specific positions. For example, players occupying roles demanding greater running and sprinting capacity may develop stronger quadriceps, impacting extension performance [24,26]. Conversely, imbalances between these muscle groups, such as reduced hamstring strength relative to the quadriceps, have been linked to injury risk and functional deficits [22,24]. The fact that only PT extension (R) showed positional differences suggests that some roles may entail specific muscle adaptations or imbalances, emphasizing the importance of tailored conditioning programs and injury prevention strategies that consider the functional roles of each position in the game [4,33]. Ruas et al. [32] detailed the variation in body composition and strength profiles according to playing position, reinforcing the notion that different positional roles require specific physical attributes. In their paper, Paillard [34] explored how dynamic balance was affected by sport-specific movements, particularly in positions such as midfield and forward, where frequent, rapid changes in direction are common. This work provides context for the discussion of methodological differences in balance assessments. This review synthesizes research on hamstring-to-quadriceps strength imbalances and their relationship to injury risk. The recommendations and findings discussed by Croisier et al. [22] support the assertion that imbalances in these ratios are commonly linked with a higher incidence of muscle strains, underscoring the importance of balanced strength training. Importantly, the absence of statistically significant differences for most variables should not be interpreted as definitive evidence of positional equivalence. With only 4–5 players per position, this study was substantially underpowered to detect negligible to small-medium effect sizes that may nonetheless be practically meaningful for training prescription and injury risk management. Future research with larger sample sizes (recommended n = 15–20 per position based on a priori power calculations) is essential to clarify whether the observed trends in body composition and functional performance represent true positional adaptations.

The positional differences observed in this study should be interpreted within the framework of the existing literature describing the neuromuscular and postural demands associated with specific roles in football. Also, this study’s expectations stemmed from the distinct physical and tactical demands associated with each specific playing position. CBs and FBs exhibited patterns consistent with higher absolute torque values, primarily explained by their greater body mass and height. The strongest correlations were observed between body mass and muscle torque, indicating that heavier players generated greater force in both extensors and flexors. Height was also positively associated with torque performance, suggesting that taller players benefit from greater leverage and cross-sectional muscle area. As expected, the H:Q ratios showed an inverse relationship with extensor torque, reflecting a quadriceps-dominant profile among these positions. Age was not a significant predictor of strength outcomes, supporting the notion that morphological and functional characteristics, rather than chronological age, determine torque performance in this cohort. In contrast, CMs demonstrated the most symmetrical limb performance, prioritizing interlimb balance over maximal strength output. WMs and Fs revealed higher hamstring dominance and H:Q ratios, which align with the neuromuscular requirements for rapid accelerations and decelerations during offensive transitions. Previous studies have demonstrated that defenders typically exhibit greater absolute strength and stability-related adaptations due to frequent involvement in aerial duels and body contact situations, whereas midfielders display enhanced dynamic balance and endurance capacities given their continuous involvement in transitional phases of play. Conversely, forwards and wide players often rely more heavily on explosive strength and rapid changes of direction to execute high-intensity actions in attacking sequences. Recent evidence supports these functional distinctions, such as Loturco et al. [33], who reported that sprint and power-related neuromuscular characteristics varied markedly across playing positions, while Vieira et al. [35] identified position-specific postural control strategies during both static and dynamic balance tasks in professional players.

Overall, these findings indicate clear positional adaptations in muscle balance and torque expression, reflecting the distinct biomechanical and tactical demands of each role on the pitch. The current study applied validated measurement tools, including the UBT and isokinetic dynamometer, which ensured reliable assessments of dynamic postural control and muscle performance. Additionally, by focusing on professional football players, the study provides valuable insights applicable to elite football contexts. However, certain limitations include the small sample size, based on just one, and was unevenly distributed across positions, particularly for forwards, which may limit the generalizability of results. The small sample size and uneven group distribution (4–5 players per position) considerably restrict the statistical power to detect meaningful between-group differences. Although the non-parametric approach is appropriate for small samples and non-normally distributed data, the limited number of participants per positional group (<10) increases the risk of Type II error, potentially obscuring true differences of practical importance. Post hoc power analysis indicated that the study achieved approximately 25–30% power to detect negligible to medium effect sizes for the main outcome (peak torque extension), suggesting that several non-significant findings may reflect insufficient sensitivity rather than the absence of positional effects [25]. Consequently, the results should be interpreted with caution, emphasizing effect size magnitude and practical trends rather than statistical significance alone. Future research should include larger and more balanced samples (n ≥ 15–20 players per position) to enhance inferential validity, allow for multivariate modeling, and better characterize positional differences in neuromuscular and postural control performance. The cross-sectional design also precludes longitudinal analysis in future perspectives, preventing insights into how attributes evolve over a season and different season phases. Also, future research should extend the analysis to other teams and competition contexts. Finally, the low internal consistency indicates potential variability in the measurements.

## 5. Conclusions

This study compared anthropometric, functional, and neuromuscular profiles of professional football players across field positions. Although no significant positional differences were found, small-to-moderate effect sizes suggest underlying trends related to specific positional demands. The limited sample size and statistical power constrain interpretation, emphasizing that non-significant findings should not be viewed as evidence of equivalence. Practically, individualized training focused on hamstring–quadriceps balance and dynamic postural control is recommended to enhance performance and reduce injury risk.

## Figures and Tables

**Figure 1 jfmk-10-00447-f001:**
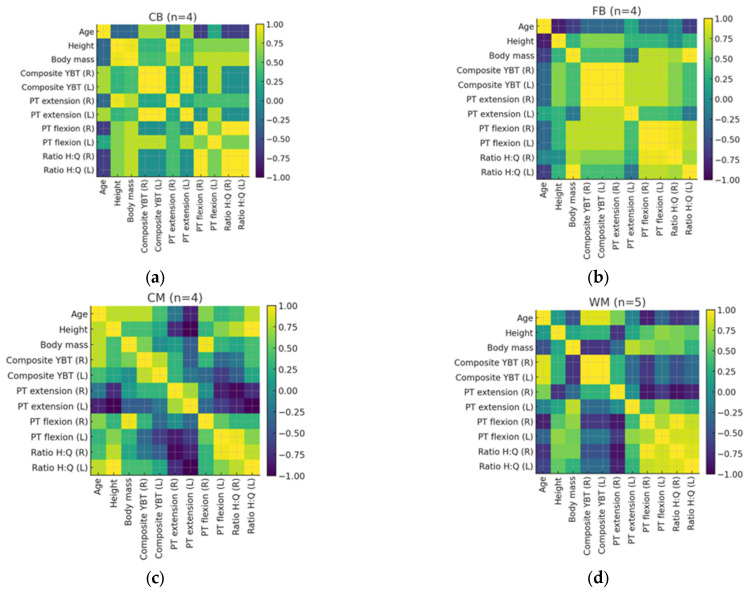
Spearman correlations matrix between variables according to field positions: (**a**) quadrant 1 (central back), (**b**) quadrant 2 (full-back); (**c**) quadrant 3 (central midfield); (**d**) quadrant 4 (wide midfield), (**e**) quadrant 5 (forward), (**f**) quadrant 6 (total). The correlation magnitude was classified as: trivial if r ≤ 0.1, small if r = 0.1–0.3, moderate if r = 0.3–0.5, large if r = 0.5–0.7, and very large if r = 0.7–0.9 and almost perfect if r ≥ 0.9. Abbreviations: Composite YBT (R)—Composite YBT score for the right side, combining multiple performance metrics; Composite YBT (L)—Composite YBT score for the left side, combining multiple performance metrics; CV—Coefficient of variation; PT_extension_ (R)—Peak torque during right-side extension (measured in Newton meters); PT_extension_ (L)—Peak torque during left-side extension (measured in Newton meters); PT_flexion_ (R)—Peak torque during right-side flexion (measured in Newton meters); PT_flexion_ (L)—Peak torque during left-side flexion (measured in Newton meters); Ratio H:Q (L)—Ratio of hamstring to quadriceps strength on the right side; Ratio H:Q (R)—Ratio of hamstring to quadriceps strength on the left side.

**Table 1 jfmk-10-00447-t001:** Descriptive statistics of detailed characteristics of the team.

Variable	CB (n = 4)	FB (n = 4)	CM (n = 4)	WM (n = 5)	FW (n = 4)	Total (n = 21)
Age	26.8 ± 4.5	26.5 ± 2.4	23.0 ± 3.9	20.4 ± 2.1	24.8 ± 3.0	24.1 ± 3.8
Height	185.8 ± 6.7	184.5 ± 1.7	181.5 ± 5.1	181.2 ± 5.0	183.5 ± 9.9	183.2 ± 5.8
Body Mass	80.1 ± 5.1	80.5 ± 3.0	71.40 ± 4.86	75.3 ± 3.4	80.8 ± 4.5	77.4 ± 5.3

**Table 2 jfmk-10-00447-t002:** Descriptive statistics for body composition, dynamic postural control (right and lefts CS), and isokinetic muscle strength (peak torque and ratios) between different field positions.

Variable	Position	N	Average	Average (Min–Max)	CV
Age (y)	CB	4	26.8 ± 4.50	19.0–32.0	0.17
	FB	4	26.5 ± 2.38	23.0–28.0	0.09
	CM	4	23.0 ± 3.92	18.0–23.0	0.17
	WM	5	20.4 ± 2.07	22.0–27.0	0.10
	FW	4	24.8 ± 2.99	21.0–28.0	0.12
	Total	21	24.1 ± 3.83	18.0–32.0	0.16
Height (cm)	CB	4	185.8 ± 6.65	176.0–191.0	0.04
	FB	4	184.5 ± 1.73	183.0–187.0	0.01
	CM	4	181.5 ± 5.07	175.0–183.0	0.03
	WM	5	181.2 ± 5.00	177.8–188.0	0.03
	FW	4	183.5 ± 9.88	169.0–191.0	0.05
	Total	21	183.2 ± 5.82	169.0–193.0	0.03
Body mass (kg)	CB	4	80.0 ± 5.13	73.8–86.2	0.06
	FB	4	80.4 ± 3.04	77.7–84.8	0.04
	CM	4	71.4 ± 4.86	65.7–74.5	0.07
	WM	5	75.3 ± 3.42	74.7–76.1	0.05
	FW	4	80.8 ± 4.47	74.4–84.6	0.06
	Total	21	77.5 ± 5.29	65.7–89.6	0.07
Composite YBT (R)	CB	4	90.0 ± 7.54	82.3–100.0	0.08
	FB	4	92.1 ± 5.26	85.3–96.7	0.06
	CM	4	89.6 ± 12.74	76.0–105.4	0.14
	WM	5	98.3 ± 8.90	95.4–104.4	0.09
	FW	4	84.2 ± 13.73	75.2–104.6	0.16
	Total	21	91.2 ± 10.23	75.2–105.4	0.11
Composite YBT (L)	CB	4	95.0 ± 1.53	93.1–96.6	0.02
	FB	4	89.7 ± 6.11	81.8–95.1	0.07
	CM	4	90.6 ± 10.60	80.9–105.9	0.12
	WM	5	96.5 ± 8.50	86.7–105.6	0.09
	FW	4	83.5 ± 9.06	75.70–96.5	0.11
	Total	21	91.3 ± 8.48	75.7–105.9	0.09
PT extension (R)	CB	4	257.5 ± 25.96	220.0–292.0	0.10
	FB	4	262.8 ± 12.45	247.0–275.0	0.05
	CM	4	215.2 ± 25.29	194.0–241.0	0.12
	WM	5	235.0 ± 16.31	194.0–260.0	0.07
	FW	4	225.0 ± 27.26	188.0–247.0	0.12
	Total	21	238.9 ± 26.87	188.0–311.0	0.11
PT extension (L)	CB	4	251.5 ± 21.86	226.0–279.0	0.09
	FB	4	263.0 ± 37.73	226.0–297.0	0.14
	CM	4	205.5 ± 24.77	190.0–240.0	0.12
	WM	5	244.6 ± 25.04	182.0–273.0	0.10
	FW	4	233.5 ± 20.57	205.0–254.0	0.09
	Total	21	239.9 ± 30.82	182.0–355.0	0.13
PT flexion (R)	CB	4	151.8 ± 22.32	126.0–176.0	0.15
	FB	4	155.5 ± 30.82	123.0–187.0	0.20
	CM	4	134.2 ± 1.26	100.0–145.0	0.01
	WM	5	139.8 ± 27.36	133.0–136.0	0.20
	FW	4	155.2 ± 15.82	134.0–170.0	0.10
	Total	21	147.0 ± 21.97	100.0–187.0	0.15
PT flexion (L)	CB	4	151.0 ± 15.81	132.0–171.0	0.10
	FB	4	156.0 ± 31.60	123.0–199.0	0.20
	CM	4	142.8 ± 5.68	100.0–171.0	0.04
	WM	5	144.0 ± 29.38	138.0–151.0	0.20
	FW	4	138.2 ± 9.32	126.0–146.0	0.07
	Total	21	146.3 ± 20.44	100.0–199.0	0.14
H:Q ratio (R)	CB	4	58.92 ± 6.68	55.0–80.0	11.34
	FB	4	59.05 ± 10.34	49.8–68.0	17.51
	CM	4	61.46 ± 11.39	41.5–68.6	18.53
	WM	5	69.25 ± 4.89	51.2–70.1	7.06
	FW	4	54.15 ± 3.89	62.7–74.2	7.18
	Total	21	61.32 ± 9.48	41.5–80.0	15.46
H:Q ratio (R)	CB	4	60.78 ± 3.2	57.1–64.4	5.32
	FB	4	59.60 ± 8.20	50.2–67.7	13.76
	CM	4	62.32 ± 11.22	45.9–74.2	18.01
	WM	5	66.47 ± 15.77	51.6–83.0	23.72
	FW	4	59.35 ± 2.72	56.9–61.9	4.58
	Total	21	60.78 ± 3.23	43.1–83.0	5.32

Abbreviations: Composite YBT (R)—Composite YBT score for the right side, combining multiple performance metrics; Composite YBT (L)—Composite YBT score for the left side, combining multiple performance metrics; CV—Coefficient of variation; PT_extension_ (R)—Peak torque during right-side extension (measured in Newton meters); PT_extension_ (L)—Peak torque during left-side extension (measured in Newton meters); PT_flexion_ (R)—Peak torque during right-side flexion (measured in Newton meters); PT_flexion_ (L)—Peak torque during left-side flexion (measured in Newton meters); Ratio H:Q (L)—Ratio of hamstring to quadriceps strength on the right side; Ratio H:Q (R)—Ratio of hamstring to quadriceps strength on the left side. Note: CVs of 5% or less were considered good performance, whereas CVs of 10% were described as bad performance.

**Table 3 jfmk-10-00447-t003:** Non-parametric analysis (Kruskal–Wallis test) of positional differences for body composition, dynamic postural control, and isokinetic muscle strength of the lower limbs.

Variable	H	*p* Value	η^2^	Interpretation
Age (y)	9.00	0.061	0.31	Large
Height (cm)	3.21	0.523	0.01	Negligible
Body mass (kg)	7.83	0.098	0.24	Large
Composite YBT (R)	4.67	0.323	0.04	Negligible
Composite YBT (L)	5.61	0.231	0.10	Small–Medium
PT extension (R)	9.84	0.043	0.37	Large
PT extension (L)	6.68	0.154	0.17	Large
PT flexion (R)	2.85	0.583	0.01	Negligible
PT flexion (L)	1.77	0.778	0.03	Negligible
H:Q ratio (R)	4.81	0.308	0.05	Negligible
H:Q ratio (L)	0.99	0.911	0.00	Negligible

Abbreviations: Composite YBT R—Composite YBT score for the right side, combining multiple performance metrics; Composite YBT L—Composite YBT score for the left side, combining multiple performance metrics; CV—Coefficient of variation; PT_extension_ (R)—Peak torque during right-side extension (measured in Newton meters); PT_extension_ (L)—Peak torque during left-side extension (measured in Newton meters); PT_flexion_ (R)—Peak torque during right-side flexion (measured in Newton meters); PT_flexion_ (L)—Peak torque during left-side flexion (measured in Newton meters); Ratio H:Q (L)—Ratio of hamstring to quadriceps strength on the right side; Ratio H:Q (R)—Ratio of hamstring to quadriceps strength on the left side. A Pearson correlation matrix was computed to analyze the levels of association between body composition, dynamic postural control, and isokinetic muscle strength of the lower limbs for each field position and overall positions; Positive correlations are represented in lighter green shades, while negative correlations appear in darker tones, with color intensity indicating the strength of the association.

## Data Availability

The original contributions presented in this study are included in the article. Further inquiries can be directed to the corresponding author.

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
