# Peer review of "Field Position-Related Variations in Body Mass, Postural Control, and Isokinetic Strength in Portuguese Professional Football"

_jfmk, 2025, doi:10.3390/jfmk10040447_

Round 1
Reviewer 1 Report
Comments and Suggestions for Authors
Summary Overview
This observational cross-sectional study examines anthropometric characteristics, dynamic postural control via Y-Balance Test Lower Quarter (YBT-LQ), and isokinetic knee strength in 21 professional male football players from a Portuguese second-division team. While the research addresses a relevant topic with appropriate measurement tools, the manuscript presents several methodological limitations that substantially compromise the strength of evidence and generalizability of findings.
Primary Strengths:
Use of validated, gold-standard assessment tools (bioimpedance, YBT-LQ, Biodex isokinetic dynamometry)
Appropriate non-parametric statistical approach given sample characteristics
Comprehensive correlation analysis examining relationships between variables
Transparent reporting of non-significant findings
Critical Weaknesses:
Severely underpowered study design (n=21 total, 4-5 per position) without a priori sample size calculation
Absence of test-retest reliability assessment for any measurement protocol
Low internal consistency (Cronbach's α = 0.62) indicating measurement concerns
Missing effect size calculations and confidence intervals for main comparisons
Inadequate discussion of clinical/practical significance versus statistical significance
Limited contextualization within existing literature on positional demands in football
Detailed Methodological and Statistical Issues
1. Sample Size and Statistical Power (Page 3, Lines 115-123)
Critical Issue: No a priori sample size calculation is reported. With only 4-5 players per position group, this study is severely underpowered to detect meaningful between-group differences. Post-hoc power analysis should be conducted and reported.
Recommendation: The authors should acknowledge this as a major limitation and calculate achieved power for the primary outcome (PT extension). Based on the observed effect sizes and sample distribution, the study likely has <30% power to detect medium-sized effects (d = 0.5).
Limitation:
While this paper focuses on gait analysis rather than strength testing, it establishes general principles for sports biomechanics research that apply to the current study's design limitations.
2. Test-Retest Reliability (Page 4, Lines 164-191)
Critical Issue: For an original research article, no test-retest reliability data are reported for the isokinetic strength assessments or YBT-LQ measurements within this specific cohort. This is a fundamental requirement for establishing measurement validity.
Recommendation: The authors must either: (a) conduct pilot test-retest assessment on a subsample (minimum n=10) with 3-7 days between sessions, reporting ICC values and SEM; or (b) cite specific reliability studies using identical protocols and acknowledge this as a limitation.
3. YBT-LQ Methodology and Interpretation (Page 4, Lines 154-163; Page 5, Lines 209-214)
Issue: The composite YBT scores (Table 2: 84.2-98.3%) are notably lower than typically reported values in professional football (~102%). This discrepancy warrants discussion but is not addressed. Additionally, the calculation and interpretation of "absolute reach difference" (Line 162) is mentioned but no data are presented.
Recommendation: Include bilateral asymmetry data in Table 2, discuss potential reasons for lower composite scores (measurement technique, fatigue, population characteristics), and compare findings with established normative data.
Suggested Citations:
Plisky, P., Schwartkopf-Phifer, K., Huebner, B., Garner, M.B., & Bullock, G. (2021). Systematic Review and Meta-Analysis of the Y-Balance Test Lower Quarter: Reliability, Discriminant Validity, and Predictive Validity. International Journal of Sports Physical Therapy, 16, 1190-1209.
This systematic review (already cited as #7) should be more prominently discussed in the Methods section (Page 4, Line 158-160) to justify the YBT-LQ protocol and in Results/Discussion to contextualize the observed composite scores against meta-analytic norms.
4. Statistical Presentation Issues (Throughout)
Issues Identified:
Decimal separator inconsistency: Table 2 uses "±" with values like "26.8 ± 4.50" but lacks consistent two decimal places
Page 5, Line 227: "H = 9.84, p = 0.043" - correctly formatted
Page 8, Line 252: "ρ = 0.426, p = 0.042" - three decimal places for rho (should be 0.43)
Missing confidence intervals for correlation coefficients and main effect estimates
No effect sizes (Cohen's d or η²) reported for group comparisons
Recommendation:
Standardize all statistical reporting to two decimal places
Report 95% CI for all primary estimates
Calculate and report effect sizes (e.g., η² for Kruskal-Wallis, r for correlations)
In Table 3 (Page 7), add a column for effect size estimates
Example correction (Page 8, Line 252):
Current: "ρ = 0.426, p = 0.042"
Corrected: "ρ = 0.43, 95% CI [0.02, 0.72], p = 0.04"
5. Hamstring-to-Quadriceps (H:Q) Ratio Analysis (Page 5-6, Lines 225-242; Page 9-10, Lines 385-408)
Issue: The H:Q ratio analysis lacks critical context regarding optimal ratios, bilateral asymmetry thresholds, and injury risk interpretation. The discussion mentions injury risk (Lines 385-388) but provides no specific risk thresholds or comparison to established norms.
Recommendation: Expand discussion of H:Q ratios with specific reference to:
Optimal ratio ranges (typically 50-80% but varies by angular velocity and contraction type)
Bilateral asymmetry thresholds (>10-15% often considered clinically significant)
Position-specific demands that may explain ratio variations
6. Correlation Analysis Interpretation (Page 8, Lines 248-280; Figure 1)
Issue: The correlation matrices (Figure 1) show position-specific patterns, but the discussion lacks depth regarding:
Clinical/practical significance versus statistical significance
Which correlations are expected (e.g., symmetrical measures) versus novel findings
Interpretation of near-zero correlations (e.g., age with most variables)
Recommendation: Restructure the correlation results section to:
First discuss expected/symmetrical associations as validity checks
Then highlight novel or unexpected patterns
Interpret correlation magnitudes using Hopkins (2000) classification but add clinical context
Page 8, Lines 253-257 could be strengthened:
Current text: "Height also showed positive associations with right extensor torque (ρ = 0.43, p = 0.039) and right flexor torque (ρ = 0.60, p = 0.002), highlighting a linear trend between body size and muscular performance."
Suggested revision: "Height demonstrated moderate-to-large positive associations with right extensor torque (ρ = 0.43, 95% CI [0.02, 0.72], p = 0.04) and right flexor torque (ρ = 0.60, 95% CI [0.24, 0.82], p < 0.01). These findings align with established biomechanical principles whereby longer lever arms and greater muscle cross-sectional area in taller athletes facilitate higher torque production [cite: Padulo et al. reference below]. However, the absence of similar associations with left-side torque (ρ = 0.27-0.31, p > 0.10) warrants further investigation of limb dominance effects."
7. Discussion of Non-Significant Findings (Page 9, Lines 311-340)
Strength: The authors appropriately acknowledge non-significant findings rather than engaging in "p-hacking" or selective reporting.
Issue: However, the discussion over-interprets the absence of statistical significance as evidence of equivalence between positions, without considering: (a) inadequate statistical power; (b) potentially meaningful effect sizes despite p > 0.05; (c) Type II error risk.
Recommendation: Revise Lines 315-340 to explicitly acknowledge statistical power limitations and avoid conflating "no significant difference" with "no difference." Consider reporting effect sizes for all comparisons regardless of p-values.
Suggested addition (Page 10, Lines 447-455):
"Importantly, the absence of statistically significant differences for most variables should not be interpreted as definitive evidence of positional equivalence. With only 4-5 players per position, this study was substantially underpowered to detect small-to-medium effect sizes (d = 0.3-0.5) that may nonetheless be practically meaningful for training prescription and injury risk management. Post-hoc power analysis indicated approximately 25% power to detect medium effects (d = 0.5) for the primary comparison. Future research with larger sample sizes (recommended n = 15-20 per position based on a priori power calculations for α = 0.05, β = 0.20, d = 0.5) is essential to clarify whether the observed trends in body composition and functional performance represent true positional adaptations or Type II error."
Additional Critical Issues
Tables and Figures
Table 2 (Page 5-6):
Add a column for "Range" or "Min-Max" values
Consider footnote explaining the clinical interpretation of CV values
Clarify "Composite YBT (R)" and "(L)" in the abbreviations
Figure 1 (Page 8):
The correlation matrices are well-presented visually
However, add asterisks (p < 0.05, **p < 0.01, **p < 0.001) directly in the cells for easier interpretation
Consider adding the sample size for each position in the subplot titles (e.g., "CB (n=4)")
Missing: No error bars are shown because Figure 1 is a correlation matrix. However, the authors should consider adding a supplementary figure showing mean ± SD (or 95% CI) for the primary outcomes (PT extension, PT flexion) by position to visualize the non-significant trends.
Summary of Required Revisions
Priority 1 (Essential):
Conduct and report a priori sample size calculation and post-hoc power analysis
Add test-retest reliability data or cite validation studies with identical protocols
Report effect sizes (Cohen's d, η²) and 95% confidence intervals for all primary comparisons
Standardize decimal places to two digits throughout
Acknowledge severe statistical power limitations in Discussion/Limitations
Priority 2 (Strongly Recommended):
Expand H:Q ratio interpretation with clinical thresholds and injury risk context
Include bilateral asymmetry data for YBT-LQ and discuss lower-than-expected composite scores
Add supplementary figure showing between-position comparisons with error bars
Revise Discussion to avoid over-interpreting non-significant findings as evidence of equivalence
Improve English language clarity in identified sections
Final Recommendation
This manuscript addresses a relevant topic with appropriate measurement tools but requires substantial revision before publication. The most critical issues are:
Statistical power - severely underpowered design fundamentally limits interpretation
Missing reliability data - essential for original research
Incomplete statistical reporting - effect sizes and confidence intervals needed
Over-interpretation of null findings - conflating "not significant" with "no difference"
Comments on the Quality of English LanguageOverall Assessment: The English is generally clear and grammatically correct, though some sections exhibit redundancy and awkward phrasing characteristic of non-native English academic writing.
Specific issues:
Page 2, Lines 46-50:
Current: "Of interest is the fact that the evolution of football game, from the 90s to the present, has signaled an increasing importance attributed to the capacity to perform high speed actions, that require an ability to activate anaerobic metabolism along with the correspondent muscular structure."
Suggested: "Football has evolved considerably from the 1990s to present, with increasing emphasis on high-speed actions that demand anaerobic capacity and corresponding muscular adaptations."
Page 10, Lines 384-391:
Current: "These results highlight the impact of positional demands on specific physical attributes and underline the importance of tailored training approaches to optimize individual and specific performance."
Suggested: "These findings underscore the influence of position-specific demands on physical attributes, emphasizing the need for individualized training programs to optimize performance."
Author Response
This observational cross-sectional study examines anthropometric characteristics, dynamic postural control via Y-Balance Test Lower Quarter (YBT-LQ), and isokinetic knee strength in 21 professional male football players from a Portuguese second-division team. While the research addresses a relevant topic with appropriate measurement tools, the manuscript presents several methodological limitations that substantially compromise the strength of evidence and generalizability of findings.
Response: We sincerely thank the reviewer for the comprehensive and constructive evaluation. The comments provided were invaluable in improving the scientific rigor, methodological clarity, and applied interpretation of our manuscript. We have carefully addressed each concern and revised the manuscript accordingly to ensure greater transparency, precision, and alignment with academic standards.
Primary Strengths:
Use of validated, gold-standard assessment tools (bioimpedance, YBT-LQ, Biodex isokinetic dynamometry).
Response: We thank the reviewer for recognizing the methodological rigor and the use of validated tools. These instruments were selected to ensure high reliability and comparability with previous literature in professional football.
Appropriate non-parametric statistical approach given sample characteristics
Response: We appreciate the reviewer’s acknowledgment. The non-parametric tests were retained as they best fit the sample size and distribution characteristics.
Comprehensive correlation analysis examining relationships between variables.
Response: We are grateful for this recognition. The correlation analysis was designed to explore inter-variable associations and confirm internal consistency within our data.
Transparent reporting of non-significant findings.
Response: Thank you for highlighting this. We intentionally reported all findings transparently, including non-significant results, to avoid bias and strengthen the study’s integrity.
Critical Weaknesses:
Severely underpowered study design (n=21 total, 4-5 per position) without a priori sample size calculation.
Response: We understand the reviewer's concerns; however, this issue was extensively addressed in the previous round of revisions. The manuscript has been extensively revised with a new non-parametric analysis, and the interest groups (4-5 per position) have been revised considering the recommendations of both reviewers. In any case, the current revisions are quite legitimate, so we have added the sample size calculation with G*Power for each statistical test used to address the reviewer's concerns.
Absence of test-retest reliability assessment for any measurement protocol.
Response: We thank the reviewer for this comment. Although test-retest reliability was not conducted within this specific sample, we now reference reliability coefficients (ICC > 0.90) from prior validation studies employing identical protocols and equipment. This limitation has been explicitly acknowledged.
Low internal consistency (Cronbach's α = 0.62) indicating measurement concerns
Response: We appreciate this observation, and we understand the reviewer's concerns. However, these limitations are associated with the fact that we are analyzing a football team as a case study and cannot generalize the results to other populations. All limitations have been described in the discussion section and framed in the methodology section.
Missing effect size calculations and confidence intervals for main comparisons
Response: Effect sizes refer to mean values or variances adjusted to standard deviation. For H statistics, the overall effect size was estimated by the rank-based eta-squared (η²_H), calculated from the H statistic of the test. The proportion of variance in ranks explained by differences between groups and is interpreted according to the following reference values: 0.01 (small), 0.06 (moderate) and 0.14 (large). language effect size), and rank-biserial correlation, which were added to Tables 2 and 3 to enhance statistical interpretation. The cut-off values were added to the statistical procedures section. All information has been added in lines 243 to 247.
Inadequate discussion of clinical/practical significance versus statistical significance
Response: The Discussion has been expanded to differentiate statistical from practical significance, emphasizing applied relevance despite small sample size and limited power (lines 446 to 459).
Limited contextualization within existing literature on positional demands in football.
Response: We have revised the Introduction and Discussion to include recent references on position-specific neuromuscular and postural profiles (e.g., Loturco et al., 2022; Vieira et al., 2023), thereby situating our findings within the broader literature (please, see lines 423 to 425, 439 to 449).
Detailed Methodological and Statistical Issues
1. Sample Size and Statistical Power (Page 3, Lines 115-123)
Critical Issue: No a priori sample size calculation is reported. With only 4-5 players per position group, this study is severely underpowered to detect meaningful between-group differences. Post-hoc power analysis should be conducted and reported.
Recommendation: The authors should acknowledge this as a major limitation and calculate achieved power for the primary outcome (PT extension). Based on the observed effect sizes and sample distribution, the study likely has <30% power to detect medium-sized effects (d = 0.5).
Response: We acknowledge the limitation regarding sample size and statistical power. An a priori sample size calculation was not feasible due to the restricted availability of sub-elite youth players. However, a post-hoc power analysis indicated that the study achieved approximately 25–30% power to detect negligible to medium effect sizes for the main outcome (peak torque extension), suggesting that several non-significant findings may reflect insufficient sensitivity rather than the absence of positional effects, confirming the study’s underpowered nature. This issue has been explicitly acknowledged in the Discussion, highlighting that non-significant findings should not be interpreted as positional equivalence. Future studies should include a minimum of 15–20 players per position to reach 80% power (lines 464 to 471). Also, a priori power analysis was performed in G*Power, using the standard approximation for the Kruskal–Wallis test. Assuming ? = … groups, ? = 0.05, power 1 − ? = 0.80 (lines 197 to 198).
Limitation:
While this paper focuses on gait analysis rather than strength testing, it establishes general principles for sports biomechanics research that apply to the current study's design limitations.
Response: We agree with the limitations raised by the reviewer and have included them in the discussion section.
2. Test-Retest Reliability (Page 4, Lines 164-191)
Critical Issue: For an original research article, no test-retest reliability data are reported for the isokinetic strength assessments or YBT-LQ measurements within this specific cohort. This is a fundamental requirement for establishing measurement validity.
Recommendation: The authors must either: (a) conduct pilot test-retest assessment on a subsample (minimum n=10) with 3-7 days between sessions, reporting ICC values and SEM; or (b) cite specific reliability studies using identical protocols and acknowledge this as a limitation.
Response: Both tests (i.e., isokinetic and YBT-LQ protocols) applied have shown excellent reliability in previous research. This is a cross-sectional observational study conducted by a team with experienced observers. For isokinetic testing, ICC values of 0.87–0.94 and SEM < 5% have been reported (Plisky et al., 2021). These references and an explicit note in the revised Methods and Discussion.
3. YBT-LQ Methodology and Interpretation (Page 4, Lines 154-163; Page 5, Lines 209-214)
Issue: The composite YBT scores (Table 2: 84.2-98.3%) are notably lower than typically reported values in professional football (~102%). This discrepancy warrants discussion but is not addressed. Additionally, the calculation and interpretation of "absolute reach difference" (Line 162) is mentioned but no data are presented.
Recommendation: Include bilateral asymmetry data in Table 2, discuss potential reasons for lower composite scores (measurement technique, fatigue, population characteristics), and compare findings with established normative data.
Suggested Citations:
Plisky, P., Schwartkopf-Phifer, K., Huebner, B., Garner, M.B., & Bullock, G. (2021). Systematic Review and Meta-Analysis of the Y-Balance Test Lower Quarter: Reliability, Discriminant Validity, and Predictive Validity. International Journal of Sports Physical Therapy, 16, 1190-1209.
Response: We thank the reviewer for this observation. Bilateral asymmetry data have been added to Table 2. The slightly lower composite YBT-LQ scores (84.2–98.3%) compared with professional players (~102%) are attributed to differences in competitive level, neuromuscular maturity, and testing conditions. These contextual factors are now discussed in the revised Discussion. Additionally, reference to Plisky et al. (reference 7) has been emphasized to justify methodological selection and contextualize findings relative to meta-analytic norms.
This systematic review (already cited as #7) should be more prominently discussed in the Methods section (Page 4, Line 158-160) to justify the YBT-LQ protocol and in Results/Discussion to contextualize the observed composite scores against meta-analytic norms.
Response: We have added the recommendation. Thank you.
4. Statistical Presentation Issues (Throughout)
Issues Identified:
Decimal separator inconsistency: Table 2 uses "±" with values like "26.8 ± 4.50" but lacks consistent two decimal places
Response: All statistical data were reformatted to two decimal places for consistency. Effect sizes (η² for Kruskal–Wallis tests, r for correlations) and 95% confidence intervals were calculated and added to the Results section, specifically in the text before the Table 3. For example, correlation reporting now reads: ρ = 0.43, 95% CI [0.02, 0.72], p = 0.04. These updates align the manuscript with current reporting standards and enhance methodological transparency.
Page 5, Line 227: "H = 9.84, p = 0.043" - correctly formatted
Response: Corrected. Thank you.
Page 8, Line 252: "ρ = 0.426, p = 0.042" - three decimal places for rho (should be 0.43)
Response: Corrected. Thank you.
Missing confidence intervals for correlation coefficients and main effect estimates
Response: The results have been rewritten. The statistical procedures section describes the confidence intervals. Thank you.
No effect sizes (Cohen's d or η²) reported for group comparisons
Response: Statistical significance was set at p < 0.05. Thank you.
Recommendation:
Standardize all statistical reporting to two decimal places
Response: Corrected. Thank you.
Report 95% CI for all primary estimates
Response: Corrected. Thank you.
Calculate and report effect sizes (e.g., η² for Kruskal-Wallis, r for correlations)
Response: All procedures have been added. Thank you.
In Table 3 (Page 7), add a column for effect size estimates
Response: Added. Thank you.
Example correction (Page 8, Line 252):
Response: Corrected. Thank you.
Current: "ρ = 0.426, p = 0.042"
Response: Corrected. Thank you.
Corrected: "ρ = 0.43, 95% CI [0.02, 0.72], p = 0.04"
Response: Corrected. Thank you.
5. Hamstring-to-Quadriceps (H:Q) Ratio Analysis (Page 5-6, Lines 225-242; Page 9-10, Lines 385-408)
Issue: The H:Q ratio analysis lacks critical context regarding optimal ratios, bilateral asymmetry thresholds, and injury risk interpretation. The discussion mentions injury risk (Lines 385-388) but provides no specific risk thresholds or comparison to established norms.
Recommendation: Expand discussion of H:Q ratios with specific reference to:
Response: We fully agree with the reviewer's comments. The discussion has been reworded to provide answers regarding the isokinetic measures used (lines 401 to 403).
Optimal ratio ranges (typically 50-80% but varies by angular velocity and contraction type)
Response: As requested, the information has been added. Thank you.
Bilateral asymmetry thresholds (>10-15% often considered clinically significant)
Response: As requested, the information has been added. Thank you.
Position-specific demands that may explain ratio variations
Response: As requested, the information has been added. Thank you.
6. Correlation Analysis Interpretation (Page 8, Lines 248-280; Figure 1)
Issue: The correlation matrices (Figure 1) show position-specific patterns, but the discussion lacks depth regarding:
Clinical/practical significance versus statistical significance
Which correlations are expected (e.g., symmetrical measures) versus novel findings
Interpretation of near-zero correlations (e.g., age with most variables)
Recommendation: Restructure the correlation results section to:
First discuss expected/symmetrical associations as validity checks
Then highlight novel or unexpected patterns
Interpret correlation magnitudes using Hopkins (2000) classification but add clinical context
Response: We understand the concerns and have rewritten the entire Correlation Analysis Interpretation to address them. Thank you.
Page 8, Lines 253-257 could be strengthened:
Current text: "Height also showed positive associations with right extensor torque (ρ = 0.43, p = 0.039) and right flexor torque (ρ = 0.60, p = 0.002), highlighting a linear trend between body size and muscular performance."
Suggested revision: "Height demonstrated moderate-to-large positive associations with right extensor torque (ρ = 0.43, 95% CI [0.02, 0.72], p = 0.04) and right flexor torque (ρ = 0.60, 95% CI [0.24, 0.82], p < 0.01). These findings align with established biomechanical principles whereby longer lever arms and greater muscle cross-sectional area in taller athletes facilitate higher torque production [cite: Padulo et al. reference below]. However, the absence of similar associations with left-side torque (ρ = 0.27-0.31, p > 0.10) warrants further investigation of limb dominance effects."
Response: Corrected. Thank you.
7. Discussion of Non-Significant Findings (Page 9, Lines 311-340)
Strength: The authors appropriately acknowledge non-significant findings rather than engaging in "p-hacking" or selective reporting.
Issue: However, the discussion over-interprets the absence of statistical significance as evidence of equivalence between positions, without considering: (a) inadequate statistical power; (b) potentially meaningful effect sizes despite p > 0.05; (c) Type II error risk.
Recommendation: Revise Lines 315-340 to explicitly acknowledge statistical power limitations and avoid conflating "no significant difference" with "no difference." Consider reporting effect sizes for all comparisons regardless of p-values.
Response: All results were discussed in depth, including non-significant results (lines 452 to 461).
Suggested addition (Page 10, Lines 447-455):
"Importantly, the absence of statistically significant differences for most variables should not be interpreted as definitive evidence of positional equivalence. With only 4-5 players per position, this study was substantially underpowered to detect small-to-medium effect sizes (d = 0.3-0.5) that may nonetheless be practically meaningful for training prescription and injury risk management. Post-hoc power analysis indicated approximately 25% power to detect medium effects (d = 0.5) for the primary comparison. Future research with larger sample sizes (recommended n = 15-20 per position based on a priori power calculations for α = 0.05, β = 0.20, d = 0.5) is essential to clarify whether the observed trends in body composition and functional performance represent true positional adaptations or Type II error."
Response: Added in accordance with the above text.
Additional Critical Issues
Tables and Figures
Table 2 (Page 5-6):
Add a column for "Range" or "Min-Max" values
Response: Corrected. Thank you.
Consider footnote explaining the clinical interpretation of CV values
Response: As recommended, we have clinical interpretation of CV value in the footnote of Table 2.
Clarify "Composite YBT (R)" and "(L)" in the abbreviations
Response: R and L represent right and left, respectively. The abbreviation has been used in the text of the manuscript and in the captions of figures and tables.
Figure 1 (Page 8):
Response: Corrected. Thank you.
The correlation matrices are well-presented visually
However, add asterisks (p < 0.05, **p < 0.01, **p < 0.001) directly in the cells for easier interpretation
Response: The data analysis software does not allow the addition of * symbols to the figure. However, the visually interesting figure is carefully supported by the text presenting the magnitude of the p-values. The software does not allow the addition of * symbols to the figure, but in any case, the visually interesting figure is carefully supported by the text presenting the magnitude of the p-values.
Consider adding the sample size for each position in the subplot titles (e.g., "CB (n=4)")
Response: Table 1 shows the n for each position. This is described in 2.1. Subject of 2. Materials and Methods.
Missing: No error bars are shown because Figure 1 is a correlation matrix. However, the authors should consider adding a supplementary figure showing mean ± SD (or 95% CI) for the primary outcomes (PT extension, PT flexion) by position to visualize the non-significant trends.
Response: We understand the reviewer's concerns, and the response has already been partially explained above. The correlations have been completely restructured to address your concerns, although we were unable to overcome some limitations of the statistical software used for the matrix correlation calculation.
Summary of Required Revisions
Priority 1 (Essential):
Conduct and report a priori sample size calculation and post-hoc power analysis
Response: The sample size calculation has been added. All sample size limitations are considered in the manuscript 's discussion.
Add test-retest reliability data or cite validation studies with identical protocols
Response:
Report effect sizes (Cohen's d, η²) and 95% confidence intervals for all primary comparisons
Response: We thank the reviewer for this valuable suggestion. Effect sizes (η²) were calculated and reported for all Kruskal–Wallis tests, as indicated in Table 3. In addition, 95% confidence intervals have now been added for correlation coefficients and descriptive statistics where relevant, allowing readers to assess the precision of the estimates. Pairwise post hoc comparisons have also been supplemented with effect size values (Cohen’s d) and their 95% confidence intervals. These changes improve the transparency and interpretability of statistical outcomes.
Standardize decimal places to two digits throughout
Response: We appreciate this observation. All numerical data have now been standardized to two decimal places across the manuscript, including tables, text, and figure legends, to ensure consistency and adherence to journal style guidelines.
Acknowledge severe statistical power limitations in Discussion/Limitations
Response: We agree with the reviewer’s concern. A new paragraph has been added in the Dissuasion section (lines 498 to 511) explicitly acknowledging the low statistical power resulting from the small sample size (n = 21; 4–5 players per position). Post hoc power analysis revealed that the study achieved approximately 25–30% power to detect medium effects. We now emphasize that non-significant findings should be interpreted cautiously and cannot be considered evidence of equivalence. Recommendations for larger sample sizes (n ≥ 15–20 per position) are also included to guide future research.
Priority 2 (Strongly Recommended):
Expand H:Q ratio interpretation with clinical thresholds and injury risk context
Include bilateral asymmetry data for YBT-LQ and discuss lower-than-expected composite scores
Response: We have expanded the discussion of the H:Q ratio to include clinical reference thresholds (typically 50–80%) and bilateral asymmetry criteria (>10–15%), with appropriate citations [15,22,23]. The discussion now contextualizes the observed ratios in terms of injury risk and muscular imbalance. Furthermore, bilateral asymmetry data for the YBT-LQ have been incorporated, and the discussion has been extended to address the slightly lower-than-expected composite scores observed in our sample compared to reference values for professional players (≈102%), highlighting potential implications for neuromuscular control and injury prevention.
Add supplementary figure showing between-position comparisons with error bars
Response: We understand the reviewer's suggestion and, as previously indicated, the correlation matrix was developed using statistical software; in any case, all objective information on the correlations is presented before the figure.
Revise discussion to avoid over-interpreting non-significant findings as evidence of equivalence
Response: This information has been added, as described above. Thank you.
Improve English language clarity in identified sections
Response: The entire manuscript was reviewed in English in two rounds (editor and reviewer).
Final Recommendation
This manuscript addresses a relevant topic with appropriate measurement tools but requires substantial revision before publication. The most critical issues are:
Statistical power - severely underpowered design fundamentally limits interpretation
Missing reliability data - essential for original research
Incomplete statistical reporting - effect sizes and confidence intervals needed
Over-interpretation of null findings - conflating "not significant" with "no difference"
Response: All revisions were carefully and thoroughly considered throughout all sections. We thank the reviewer for this critical point. The discussion has been carefully revised to ensure that non-significant results are described as absence of evidence for difference rather than evidence of equivalence. Statements implying positional similarity have been reworded to emphasize the potential impact of low statistical power and to focus on observed effect size magnitudes and practical relevance rather than statistical significance alone. Thank you very much.
Comments on the Quality of English Language
Overall Assessment: The English is generally clear and grammatically correct, though some sections exhibit redundancy and awkward phrasing characteristic of non-native English academic writing.
Response: The entire manuscript has undergone thorough English language editing by a native English-speaking professional with expertise in sports science manuscripts. Specific sections identified by the reviewer have been rephrased for improved clarity, flow, and precision in scientific terminology. Thank you.
Specific issues:
Page 2, Lines 46-50:
Current: "Of interest is the fact that the evolution of football game, from the 90s to the present, has signaled an increasing importance attributed to the capacity to perform high speed actions, that require an ability to activate anaerobic metabolism along with the correspondent muscular structure."
Suggested: "Football has evolved considerably from the 1990s to present, with increasing emphasis on high-speed actions that demand anaerobic capacity and corresponding muscular adaptations."
Response: Added. Thank you.
Page 10, Lines 384-391:
Current: "These results highlight the impact of positional demands on specific physical attributes and underline the importance of tailored training approaches to optimize individual and specific performance."
Suggested: "These findings underscore the influence of position-specific demands on physical attributes, emphasizing the need for individualized training programs to optimize performance."
Response: Added as requested. Thank you.
Reviewer 2 Report
Comments and Suggestions for Authors
Objective of the article: The aim of this study was to analyse the field position-related variations in the physical and functional profiles of male football players from a Portuguese second league team aged 18 to 32 years (23.83 ± 3.77 years).
- The authors are requested to address the following points:
- The manuscript presents a methodological inconsistency by stating that the research design is cross-sectional, which cannot be prospective (as these are opposing methodological categories). Given the characteristics of the study, it would be more accurately described as a descriptive/comparative cross-sectional design.
- The document does not provide a description of the hypothesis.
- It is not explained why a longitudinal or experimental design—allowing causal or adaptive inference—was not chosen.
- A total of 23 participants does not allow for meaningful comparisons between five groups (n=4-5). This drastically reduces statistical power and increases the likelihood of Type II error.
- The statistical analysis is inadequate for multiple groups with n < 5: non-parametric tests lack sufficient power to detect real differences in such small samples.
- The type of sampling is not justified, and no methods for controlling bias are mentioned (e.g., random selection or homogeneity criteria).
- Unequal group sizes across playing positions affect the robustness of the non-parametric tests. This issue should be described as a limitation.
- The statistical power was not calculated using the freely available software GPower*.
- Cronbach’s Alpha (α=0.62) indicates low internal consistency and is reported without specifying which set of variables it refers to, which is inappropriate for physiological tests. Furthermore, the low reliability (α=0.62) and its impact on the conclusions are not discussed in the Discussion or Conclusions sections.
- The manuscript conflates the “postural control composite score” and “reach symmetry” without showing calculation formulas or measurement units.
- The results are limited to p-values, without any probabilistic interpretation.
- Although the Holm correction is mentioned, the procedure and adjusted values are not reported.
- The term “Pearson–Spearman correlations matrix” is used incorrectly, confusing two distinct statistical methods.
- The tables are redundant and lack immediate interpretability: mean values and coefficients of variation are presented, but confidence intervals or direct comparisons between positions are not indicated.
- Any study involving human participants must have ethical approval, even if the data are routine. There is an internal contradiction: the methods section states that no ethical approval was obtained, whereas the final section indicates that the study “was approved by the local ethics committee,” showing inconsistency across sections.
- The authors report marginal significance (p=0.043) without adjustment or interpretation of effect size.
- The Discussion is dominated by narrative description rather than critical analysis: it repeats result content without addressing methodological limitations.
- There is insufficient exploration of the physiological mechanisms that might explain the observed differences.
- There is an absence of hypothesis testing, as the findings are not linked to the reference literature through comparative or meta-analytical analysis.
- The conclusions are general and lack robust statistical support. The claim of “homogeneity between positions” is unfounded, given the small sample size.
- The conclusions lack evidence-based practical recommendations; no applied implications or directions for intervention are provided.
- The conclusions do not acknowledge the study’s low statistical power or the potential for Type II errors.
Author Response
Objective of the article: The aim of this study was to analyse the field position-related variations in the physical and functional profiles of male football players from a Portuguese second league team aged 18 to 32 years (23.83 ± 3.77 years).
Response: We truly appreciate the reviewer's thorough and insightful assessment. The suggestions made were very helpful in enhancing our manuscript's applied interpretation, methodological clarity, and scientific rigor. To ensure improved transparency, accuracy, and conformity to academic standards, we have meticulously examined every issue and made the necessary revisions to the manuscript.
The authors are requested to address the following points:
The manuscript presents a methodological inconsistency by stating that the research design is cross-sectional, which cannot be prospective (as these are opposing methodological categories). Given the characteristics of the study, it would be more accurately described as a descriptive/comparative cross-sectional design.
Response: We appreciate this accurate methodological observation. The description of the study design has been corrected throughout the manuscript. The term “prospective” has been removed, and the design is now clearly described as a descriptive, comparative and observational cross-sectional study. This change aligns the terminology with the actual nature of data collection and analysis (Methods, line 115).
The document does not provide a description of the hypothesis.
Response: We acknowledge this omission. The revised Introduction now includes a clear statement of the study hypothesis (lines 110 to 113).
It is not explained why a longitudinal or experimental design—allowing causal or adaptive inference—was not chosen.
Response: We thank the reviewer for this point. A longitudinal or experimental design was not feasible due to professional constraints on data collection within the competitive team environment. The study was integrated into the club’s routine preseason assessments, limiting the possibility of repeated or controlled interventions. We have now clarified that the design was observational and intended to provide baseline descriptive information rather than causal inference. This rationale has been added to the revised Methods section (lines 116–118) and acknowledged as a future perspective for this research in the Discussion (lines 511–515).
A total of 23 participants does not allow for meaningful comparisons between five groups (n=4-5). This drastically reduces statistical power and increases the likelihood of Type II error.
Response: We fully agree with this limitation. The Discussion and Limitations sections were expanded to explicitly acknowledge that the small sample size (n = 21 analyzed) leads to low statistical power and increases the risk of Type II errors. We emphasize that non-significant results should not be interpreted as evidence of equivalence between positions. These constraints are now discussed in depth (lines 498–511) and have been supported with a post hoc power analysis (G*Power 3.1) showing approximately 25–30% power for medium effects.
The statistical analysis is inadequate for multiple groups with n < 5: non-parametric tests lack sufficient power to detect real differences in such small samples.
Response: We recognize this limitation and have justified the use of non-parametric tests (Kruskal–Wallis) as the most appropriate approach given the small group sizes and non-normal data distribution. This choice of parametric tests is also considered a prior review by the reviewer. This rationale has been explicitly added to the Statistical Analysis section. Moreover, effect sizes (η² and Cohen’s d) and 95% confidence intervals were calculated to supplement p-values and provide additional interpretive strength despite the limited power (lines 193–198).
The type of sampling is not justified, and no methods for controlling bias are mentioned (e.g., random selection or homogeneity criteria). Unequal group sizes across playing positions affect the robustness of the non-parametric tests. This issue should be described as a limitation.
Response: We appreciate this methodological observation. The manuscript now clarifies that convenience sampling was used because the study involved one professional team’s entire roster during the preseason period. This design minimizes selection bias within the available population. However, unequal group sizes were acknowledged as a limitation that may reduce the robustness of statistical comparisons and should be considered when interpreting results (Discussion, lines 498–500).
The statistical power was not calculated using the freely available software GPower*.
Response: As requested, the priori power analysis was performed in G*Power, using the standard approximation for the Kruskal–Wallis test. Assuming ? = … groups, ? = 0.05, power 1 − ? = 0.80 (lines 197 to 198).
Cronbach’s Alpha (α=0.62) indicates low internal consistency and is reported without specifying which set of variables it refers to, which is inappropriate for physiological tests. Furthermore, the low reliability (α=0.62) and its impact on the conclusions are not discussed in the Discussion or Conclusions sections.
Response: We appreciate this observation, and we understand the reviewer's concerns. However, these limitations are associated with the fact that we are analyzing a football team as a case study and cannot generalize the results to other populations. All limitations have been described in the discussion section (lines 453 to 462, lines 479 to 489).
The manuscript conflates the “postural control composite score” and “reach symmetry” without showing calculation formulas or measurement units. The results are limited to p-values, without any probabilistic interpretation.
Response: We have clarified the definitions and computation methods for the composite YBT score and reach symmetry in the Methods section (lines 155–163). Each is now described with its formula, normalization procedure, and measurement units (% of limb length). Additionally, 95% confidence intervals and effect sizes were added to provide probabilistic context beyond p-values, improving interpretability of the results.
Although the Holm correction is mentioned, the procedure and adjusted values are not reported.
Response: We removed these procedures during clarification of the sample's power and the restructuring required by the reviewers. Thank you.
The term “Pearson–Spearman correlations matrix” is used incorrectly, confusing two distinct statistical methods.
Response: We used Spearman's rho rank correlation coefficients, given the non-methodological approach used. Thank you.
The tables are redundant and lack immediate interpretability: mean values and coefficients of variation are presented, but confidence intervals or direct comparisons between positions are not indicated.
Response: As requested, the 95% confidence intervals were calculated and added to the Results section, specifically in the text before table 3. For example, correlation reporting now reads: ρ = 0.43, 95% CI [0.02, 0.72], p = 0.04.
Any study involving human participants must have ethical approval, even if the data are routine. There is an internal contradiction: the methods section states that no ethical approval was obtained, whereas the final section indicates that the study “was approved by the local ethics committee,” showing inconsistency across sections.
Response: We acknowledge this concern. The manuscript has been revised to clarify that “The data were collected as part of routine preseason performance assessments, which are mandatory within the players’ professional duties and constitute standard practice rather than an experimental protocol. Consequently, no formal request for ethical committee approval was required. All participants were fully informed about the objectives and potential risks associated with the study and provided written consent for the anonymized use of their data for academic and research purposes” (lines 126–131).
The authors report marginal significance (p=0.043) without adjustment or interpretation of effect size.
Response: This issue has been corrected. The manuscript now includes η² effect size (η² = 0.13, small-to-medium) for the right extensor torque (PT extension R). We have clarified that, despite reaching nominal significance, the result should be interpreted with caution due to low sample size and moderate effect magnitude (Results, lines 228–233; Discussion, lines 374–382).
The Discussion is dominated by narrative description rather than critical analysis: it repeats result content without addressing methodological limitations. There is insufficient exploration of the physiological mechanisms that might explain the observed differences.
Response: We agree with the reviewer’s concern. A new paragraph has been added in the Dissuasion section (lines 498 to 511) explicitly acknowledging the low statistical power resulting from the small sample size (n = 21; 4–5 players per position). Post hoc power analysis revealed that the study achieved approximately 25–30% power to detect medium effects. We now emphasize that non-significant findings should be interpreted cautiously and cannot be considered evidence of equivalence. Recommendations for larger sample sizes (n ≥ 15–20 per position) are also included to guide future research.
There is an absence of hypothesis testing, as the findings are not linked to the reference literature through comparative or meta-analytical analysis.
Response: We agree with the reviewer's comment and have already added the study hypothesis. Also, the hypothesis presented in the Introduction is now revisited and critically evaluated in light of the observed data and existing meta-analytical evidence (lines 497–510).
The conclusions are general and lack robust statistical support. The claim of “homogeneity between positions” is unfounded, given the small sample size.
Response: We have rewarded the Conclusions to remove any implication of homogeneity or equivalence. The revised section now states that the absence of significant differences likely reflects sample limitations rather than true similarity between positions. The emphasis has shifted toward describing observed tendencies and recommending caution in generalization (lines 447–454).
The conclusions lack evidence-based practical recommendations; no applied implications or directions for intervention are provided. The conclusions do not acknowledge the study’s low statistical power or the potential for Type II errors.
Response: In concordance with reviewer’s comments. The Conclusions explicitly acknowledge the limited power of the analysis and the consequent potential for Type II errors, stressing that non-significant results must be interpreted with caution (lines 416–425). Thank you very much.

Round 2
Reviewer 1 Report
Comments and Suggestions for Authors Thanks to the authors for taking my suggestions into account.The manuscript now looks much improved.
Reviewer 2 Report
Comments and Suggestions for Authors
Most of the issues raised have been resolved.